# Modeling Expectation Violation in Intuitive Physics with Coarse Probabilistic Object Representations

**Kevin A. Smith**[1,2,*] **Lingjie Mei**[3,*]**, Shunyu Yao**[4,*]**, Jiajun Wu**[3]
**Elizabeth Spelke**[2,5]**, Joshua B. Tenenbaum**[1,2,3]**, Tomer D. Ullman**[2,5]

[1] MIT BCS, [2] Center for Brains, Minds, & Machines, [3] MIT CSAIL,
[4] Princeton University, [5] Harvard Psychology

## Abstract

From infancy, humans have expectations about how objects will move and interact. Even young children expect objects not to move through one another, teleport, or disappear. They are surprised by mismatches between physical expectations and perceptual observations, even in unfamiliar scenes with completely novel objects. A model that exhibits human-like understanding of physics should be similarly surprised, and adjust its beliefs accordingly. We propose ADEPT, a model that uses a coarse (approximate geometry) object-centric representation for dynamic 3D scene understanding. Inference integrates deep recognition networks, extended probabilistic physical simulation, and particle filtering for forming predictions and expectations across occlusion. We also present a new test set for measuring violations of physical expectations, using a range of scenarios derived from developmental psychology. We systematically compare ADEPT, baseline models, and human expectations on this test set. ADEPT outperforms standard network architectures in discriminating physically implausible scenes, and often performs this discrimination at the same level as people.

## 1 Introduction

People have a rich understanding of everyday physics that they use to predict how the future might unfold [Battaglia et al., 2013, Smith and Vul, 2013], plan actions, and manipulate tools [Osiurak and Badets, 2016]. This commonsense reasoning includes a set of early developing or possibly innate expectations about the behavior of objects, which are part of 'core knowledge' [Spelke and Kinzler, 2007]. For example, even very young infants generally expect objects to remain coherent wholes that follow spatially contiguous paths, and not wink in and out of existence [Baillargeon, 1987, Spelke et al., 1992]. Intelligent machines that can interact with the physical world in a human-like way should hold human-like physical intuitions [Lake et al., 2017].

We propose that human-like physical understanding is based on explicit object-centric representations and their associated dynamics, similar to the idea of a Mental Game Engine [Battaglia et al., 2013, Ullman et al., 2017]. Such object representations have constant physical properties regardless of their perceptual appearance, an appearance that can differ greatly depending on factors such as viewpoint, distance, and occlusion. We assume these object-based representations as given, and propose that the main computational burden is learning how visual input is parsed into these representations.

Our work has two aims. We first aim to formalize human (especially infant) physical cognition. But we also aim to show how this modeling of infant cognition can inspire more robust AI vision systems that extract physical object representations from video and can detect violations of physical expectations to use as learning signals, similar to infants [Stahl and Feigenson, 2015].



Figure 1: Frames taken from a physically implausible video, in which a yellow cube seems to disappear behind the occluder. Agents that observe this scene should find it surprising, and use this knowledge as a guide to explore the properties of the objects or dynamics that caused the yellow cube to disappear.

We present a new model, "Approximate Derenderer, Extended Physics, and Tracking" (ADEPT; Fig. 2), that closes the loop between cognitive models of intuitive physics which assume object parses are given [e.g. Battaglia et al., 2013, Smith and Vul, 2013, Hamrick et al., 2016, Ullman et al., 2018], and computer vision models that parse images into physical object representations [Wu et al., 2017a]. Importantly, we suggest that perception does not have to be exact to capture basic physical expectations [Ullman et al., 2017]. Approximate perception allows the model to trivially extend to novel objects at a loss of object identity information; this is similar to the way young infants can reason about new objects but are insensitive to changes of object shape [Xu and Carey, 1996]. Our model (i) learns to approximately parse novel arbitrarily shaped objects into approximate geometric forms, (ii) makes extended predictions about future world states, by using a robust dynamics model that combines the predictions of a standard physics engine with with the possibility of resampling aspects of the dynamic scene from the prior, and (iii) uses a particle filter to tie together parsing and predicting, allowing it to track objects over occlusion.

In the spirit of Riochet et al. [2018] and Piloto et al. [2018], we evaluate our model using the Violation of Expectations (VoE) paradigm from developmental psychology. In this paradigm, models are shown scenes that are matched as closely as possible, except that one scene contains an event that violates intuitive physical expectations (e.g., an object disappearing; Fig 1). A model passes the test if its predictions diverge from observations more strongly in the violation video than the control.

We tested the ADEPT model on eight different scenarios, which replicate tests from developmental psychology and capture different aspects of early core object knowledge. These scenarios examine concepts such as permanence (objects do not appear or disappear for no reason), continuity (objects move along connected trajectories), and solidity (objects cannot move through one another). We compared ADEPT to models that acquire physical expectations without explicit object representations, and found that only ADPET discriminates violations from control stimuli at above chance rates in all eight scenarios, and did so at similar rates as humans.

In sum, this paper makes three contributions: (1) ADEPT, a novel, object-centric model that can discriminate physically implausible scenes at near-human levels; (2) A new stimulus set closely based on developmental psychology for probing core physics knowledge, with more differentiated training and test sets to test stronger generalization than previous such stimuli; (3) A new data set of human adult judgments on our stimuli, allowing us to investigate not just whether a model discriminates violations above chance, but whether it has human-like expectations about physical events.

## 2 Related Work

The ADEPT model relies on two modules: an 'inverse graphics' module that infers an object-centric representation from raw images, and a 'physical simulation' module that predicts future object states from current beliefs. Working within the framework of 'vision-as-inverse-graphics' [Yuille and Kersten, 2006], researchers have come up with various ways to integrate recognition nets with rendering engines [e.g, Wu et al., 2017b, Che et al., 2018, Kulkarni et al., 2015, Huang et al., 2018], in order to extract information about objects (e.g., their number and attributes) from pixel input. The ADEPT model extends this work by extracting very coarse shape information, which allows it to naturally generalize to new objects.

Recent work on physical prediction can be roughly divided into models that learn in a pure end-to-end way from raw pixels with minimal built-in structure [e.g, Mottaghi et al., 2016, Agrawal et al., 2016, Finn et al., 2016, Fragkiadaki et al., 2016, Eslami et al., 2016, 2018, Lerer et al., 2016, Vondrick et al., 2016], and models that integrate object-based representations with off-the-shelf physics engines [e.g., Battaglia et al., 2013, Wu et al., 2017a, Zheng et al., 2015, Kloss et al., 2017]. Graph-based networks lie in between these broad categories, allowing for minimal object representations and relations as nodes and edges in a graph [Battaglia et al., 2016, Chang et al., 2017, Mrowca et al., 2018]. ADEPT

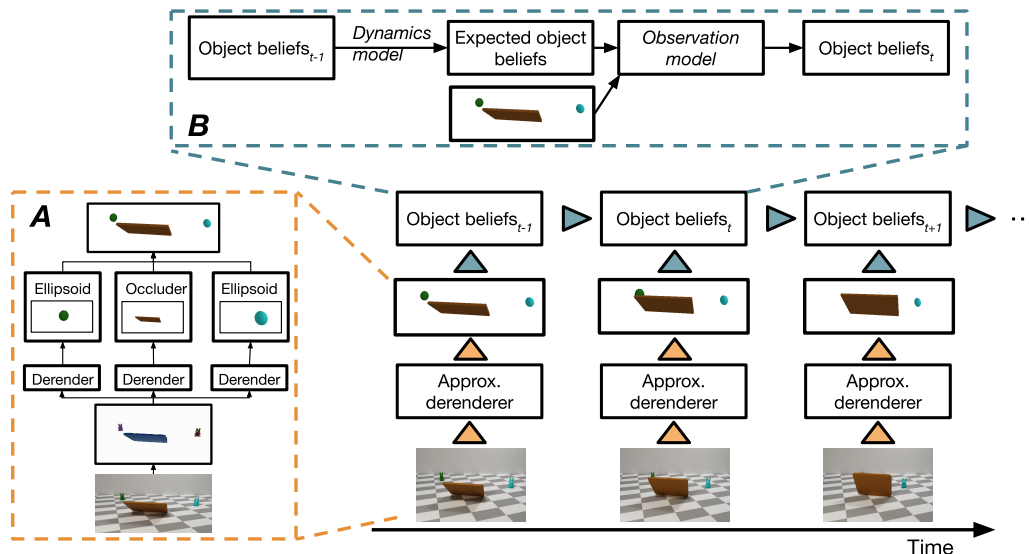

Figure 2: The ADEPT model contains two parts. **A.** The perception module segments objects, and then extracts coarse object attributes from each object segment, approximating all non-occluders as ellipsoids. **B.** The reasoning module tracks and updates beliefs based on the perception results, using the particle filter algorithm and an extended stochastic physics engine.

infers approximate object representations from visual input rather than precise geometric bodies, and integrates them with an extended physics engine that allows for robust dynamics and non-standard events. Similar to Vul et al. [2009], we represent belief distributions over object states using a particle filter to track objects and their properties even during occlusion.

Physical reasoning models are often evaluated by comparing a model's accuracy to the ground-truth unfolding of a scene, or the known physical parameters of objects. Recently, however, two groups of researchers have proposed that developmental psychology can offer bench-marks for evaluating model performance, by comparing it to the looking time patterns of young infants [Piloto et al., 2018, Riochet et al., 2018]. Our paper builds on those proposals, but differs from them in three important ways. First, the previous work used stimuli from developmental psychology in both training and testing. While the test stimuli may differ from the training in terms of object properties (a blue cube flattened by a red screen in training, but by a yellow screen in test), the training and testing are still relatively similar. In contrast, we only use developmental-psychology-like stimuli during testing. Our training videos contain a small set of objects with a relatively unconstrained motion that does not include collisions. Our test set can contain motions that are not contained in the training set, but are compositions of trained motions. Similarly, our test set includes only objects outside of the training set. This train and test split provides a stricter test of generalization, reduces the risk of overfitting, and is more human-like in that infants do not see anything like developmental psychology test stimuli in their everyday life. Second, we focus on violations of core physical principles that are observed very early in infancy (such as solidity and permance), rather than violations that are noticed only in older infants (such as changes in object shape or color [Xu and Carey, 1996], or violations of conservation of energy). Third, we evaluate and compare both pixel-based and object-based models, while prior work focuses on pixel-based methods only.

## 3 Approximate Derendering, Extended Physics, and Tracking (ADEPT)

As shown in Figure 2, the proposed Approximate Derendering, Extended Physics, and Tracking (ADEPT) model has two components: a perception module (Figure 2A) that estimates an abstract, object-centric representation $o_t$ (observation) from a raw image $x_t$ at each moment $t$, and a reasoning module (Figure 2B) that maintains a belief about the scene's physical state $p(s_t|o_{<t})$, conditioning on past observations $o_{<t}$ and using particle filtering. Further details can be found in the supplement.

### 3.1 Perception

The perception module (Figure 2A) translates a raw image $x_t$ to an object-centric presentation $o_t$ in two stages. First, $x_t$ is fed into an instance segmentation network to obtain object segments

$\hat{o}_t = \{\hat{o}_{t,i}\}_{i=1}^{n(o_t)}$, where $n(o_t)$ is the number of segments. We use each proposed object segment $\hat{o}_{t,i}$ to mask the current image $x_t$ and concatenate the masked image with original images at $x_{t-4}$, $x_{t-2}, x_t$ to account for the proposed object's visual, temporal, and contextual information. A deep convolutional feature extractor takes the concatenation and returns $o_{t,i}$, a feature vector that encodes intrinsic shape attributes (type, scale), and state attributes (location, velocity, rotation) of the object in a pre-defined format.

Our perception module approximates shapes into two types: large, thin cuboids (typically occluders), or ellipsoids of varied scales (other objects). This approximation increases robustness to novel objects and shapes, and aligns better with infant learning literature, which suggests object kind information is not used early on in spatio-temporal tracking [Xu and Carey, 1996, Ullman et al., 2017].

### 3.2 Physical Reasoning via Particle Filtering

The physics reasoning module aims to estimate $p(o_t|o_{<t})$ at each moment $t$. It maintains and updates a belief over the physical state of the scene, $s_t = \{s_{t,i}\}_{i=1}^{n(s_t)}$, where $n(s_t)$ is the number of objects, and each object state $s_{t,i}$ contains the same set of attributes as an object observation $o_{t,i}$. We assume a Hidden Markov Model (HMM) relating $s_{1...T}$ and $o_{1...T}$, whose transitional probability is governed by two models:

- **Dynamics model** $p(s_{t+1}|s_t)$. We employ a stochastic physics engine $\Theta : s_t \rightarrow s_{t+1}$ to sample object dynamics. However, standard physical dynamics will place vanishingly low probabilities on physically implausible events, and can cause degeneracy in the particle filter. We thus extend our dynamics model to include the possibility of re-sampling aspects of the dynamic scene from the prior. In the same way that any physical scene interpretation begins with an arbitrary draw from the scene prior at the beginning of a scenario, we assume that at any moment as the scene unfolds, aspects of the scene may occasionally change arbitrarily and unpredictably, which can be modeled by redrawing them from the prior, with very low probability. In principle any object or any subset of its properties can be re-sampled in this way to allow for violation of its expected dynamics, and can be handled efficiently via smart initialization such that new samples are constrained by given observations. Here we consider only re-samples of two basic object states: identity or existence (accounting for unexpected appearances or disappearances), and velocity (accounting for unexpected stopping, speeding up, or changing direction). We are also exploring more general implementations that re-sample objects' positions, shapes, colors, and other properties.

- **Observation model** $p(o_t|s_t)$. To estimate $p(o_t|s_t)$, we first match objects in our belief $\{s_{t,i}\}_{i=1}^{n(s_t)}$ with objects in the current observation $\{o_{t,i}\}_{i=1}^{n(o_t)}$, based on extrinsic attributes (color, shape, and location). We also adjust for mismatches between $s_{t,i}$ and $o_{t,i}$ in case (i) the model believes an object exists but it is not observed (e.g., due to occlusion), or (ii) an object must be added to $s_{t,i}$ because it was in a previously unobserved location (e.g., it comes out from behind an occluder). After objects are matched, the likelihood is computed based on the intersection of union (IoU) of their corresponding silhouettes (masks).

We use a particle filter to track and update beliefs. At any time $t$, ADEPT maintains a set of $M$ particles $\{s_t^{(m)}\}_{m=1}^M$ with normalized weights $\{w_t^{(m)}\}_{m=1}^M$ to approximate the belief distribution $p(s_t|o_{\leq t})$. Each step of particle filtering involves a stochastic state transition $s_t^{(m)} = \Theta(s_{t-1}^{(m)})$, re-weighting $w_t^{(m)} \propto w_{t-1}^{(m)} p(o_t|s_t^{(m)})$, and importance re-sampling based on $w_t^{(m)}$. Within this framework, our model estimates $p(o_t|o_{<t})$ in the following way:

$$p(o_t|o_{<t}) = \int_{s_t} p(s_t|o_{<t})p(o_t|s_t, o_{<t})ds_t \approx \sum_{m=1}^M w_t^{(m)} p\left(o_t \Big| s_t^{(m)}\right). \tag{1}$$

## 4 Modeling Violation of Expectation in Intuitive Physics

### 4.1 Framework

For a video $\boldsymbol{x}$ of $T$ frames $\boldsymbol{x} = x_{1...T}$, we want humans or models to output a **level of surprise** $c(\boldsymbol{x})$, a number that indicates the deviation of observations from expectations throughout $\boldsymbol{x}$. For two videos, $c(\boldsymbol{x}) > c(\boldsymbol{x}')$ implies $\boldsymbol{x}$ is more implausible than $\boldsymbol{x}'$. In humans, this measure can be obtained by explicit report (for adults) or looking time (for infants). While several instantiations of

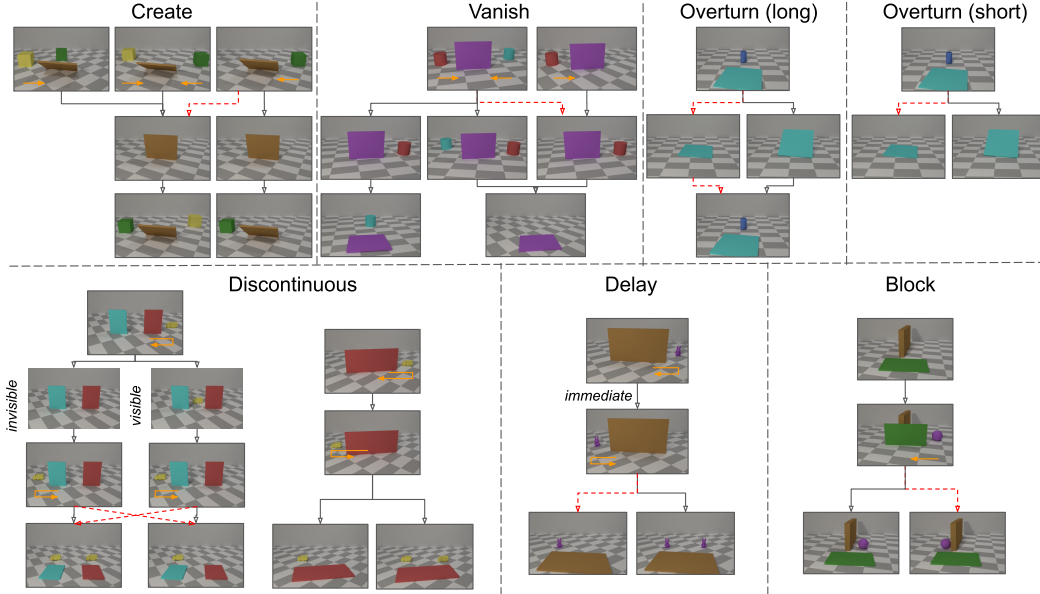

Figure 3: Diagrams of the different expectation violation scenarios. Black arrows represent physically plausible transitions between movie parts. Red dashed arrows represent transitions that violate physical expectations.

this measure are possible, for the purposes of our models here we will consider it to be the most surprising moment in a series $c(\boldsymbol{x}) = \max_t c(x, t)$, where the level of surprise at each moment $c(x_t)$ is defined according to the model. A probabilistic model like ADEPT may ground surprise as the negative log-likelihood of observations under the model $c(\boldsymbol{x}, t) \triangleq -\log p(x_t|x_{<t})$, similar to how infant measures of surprise have been found to be inversely related to the probability of an event occuring [Téglás et al., 2011, Kidd et al., 2012]. A deterministic prediction model might choose surprise to be L2 divergence between predictions and observations $c(\boldsymbol{x}, t) \triangleq ||f(x_{<t}) - x_t||_2^2$.

## 4.2 Stimuli

Inspired by developmental psychology, we designed a stimuli set to test for aspects of *core knowledge* of physics [Spelke and Kinzler, 2007]: that objects do not appear nor disappear for no reason (permanence), that objects cannot occupy the same space at the same time (solidity), and that objects move along continuous paths (continuity). These stimuli were grouped into eight scenarios, each of which included a "surprising" scene that contained a violation of physical expectation, and one to three "control" scenes that matched the surprising scenes in different aspects but did not include a violation of physical expectations (see Fig. 3, with further details and videos in the supplemental material). These scenarios included:

- **Create:** Based on Wynn [1992], this scenario tests the concept of permanence by showing an object appear from behind an occluder. This is compared to control scenes in which nothing appears, or the object was already in the scene.

- **Vanish:** This scenario is the converse of *create*, in which an object disappears while behind an occluder. This is compared to control scenes in which nothing disappears, or the object that is removed in the violation scene was never observed to begin with.

- **Overturn (long):** Based on Baillargeon et al. [1985], this scenario violates the principle of solidity by showing a screen rotate backwards and through an object that was positioned behind it, then rotates back to show the object still exists. This scenario is thus doubly surprising. In the control scenes, the screen stops rotating when it comes into contact with the object.

- **Overturn (short):** This scenario is identical to the *overturn (long)* scenario, but the video ends after the first rotation of the screen.

- **Discontinuous (invisible):** Based on Spelke et al. [1995], this scenario tests the principle of continuity by showing an object disappear behind one occluder and appear out of a spatially distinct occluder. The video ends with both occluders rotating down to show only one object, suggesting the object teleported through the intervening space. In the control scenes, two identical

|  |  | Encoder-decoder | GAN | LSTM | ADEPT |
|---|---|---|---|---|---|
| By Stimulus | Create | 0.51 [0.43, 0.58] | 0.63 [0.60, 0.66] | 0.47 [0.44, 0.51] | **0.77 [0.74, 0.80]** |
|  | Vanish | 0.52 [0.45, 0.58] | 0.50 [0.44, 0.56] | 0.50 [0.44, 0.56] | **0.83 [0.76, 0.90]** |
|  | Overturn (long) | 0.53 [0.42, 0.66] | **0.84 [0.74, 0.92]** | 0.63 [0.52, 0.74] | 0.73 [0.63, 0.82] |
|  | Overturn (short) | 0.61 [0.50, 0.73] | **0.81 [0.71, 0.90]** | 0.52 [0.39, 0.65] | **0.79 [0.70, 0.87]** |
|  | Discon. (invisible) | 0.77 [0.72, 0.82] | **0.76 [0.69, 0.82]** | 0.34 [0.25, 0.42] | **0.79 [0.72, 0.85]** |
|  | Discon. (visible) | **0.82 [0.76, 0.88]** | 0.60 [0.53, 0.66] | 0.53 [0.46, 0.6] | **0.80 [0.73, 0.86]** |
|  | Delay | 0.29 [0.19, 0.40] | 0.60 [0.48, 0.69] | 0.40 [0.32, 0.48] | **0.76 [0.68, 0.85]** |
|  | Block | 0.52 [0.40, 0.65] | 0.44 [0.32, 0.55] | 0.44 [0.35, 0.55] | **0.68 [0.57, 0.79]** |
| By Shape | Unseen shapes | 0.57 [0.53, 0.61] | 0.63 [0.59, 0.67] | 0.49 [0.45, 0.53] | **0.79 [0.76, 0.82]** |
|  | Unseen categories | **0.59 [0.51, 0.66]** | **0.61 [0.53, 0.69]** | 0.45 [0.37, 0.52] | **0.68 [0.59, 0.76]** |
|  | Geometric shapes | 0.47 [0.35, 0.58] | 0.66 [0.56, 0.77] | 0.39 [0.28, 0.49] | **0.84 [0.74, 0.94]** |
|  | Toys | 0.71 [0.58, 0.83] | 0.57 [0.53, 0.62] | 0.43 [0.35, 0.50] | **0.85 [0.72, 0.96]** |
| Average |  | 0.57 [0.54, 0.60] | 0.63 [0.60, 0.66] | 0.47 [0.44, 0.51] | **0.79 [0.72, 0.85]** |

Table 1: Relative accuracy within stimulus classes (top) and shape classes (bottom). Brackets indicate bootstrapped 95% CI. Bold indicates best performing model(s) on that category.

objects exist at the end, or there is no space between the occluders such that the object did not need to teleport.

- **Discontinuous (visible):** This scenario is similar to *discontinuous (invisible)* except that the object moves visibly between the occluders. Due to this change, the surprising scene now includes two objects, and the scene with just one object is expected (any explanation of the scene involving two objects would be less parsimonious than one object up until the moment of reveal, involving for example multiple suspicious-coincidence stop-and-start behaviors, or movement through a solid object, etc.). This is shown together with *discontinuous (invisible)* in Fig. 3 because the ends of the surprising scenes can be swapped to form a control scene for the other violation type.

- **Delay:** This scenario was designed to test the principle of continuity, by showing an object moving through an occluded area too quickly to be explained by continuous motion (involving either a sudden speed up and slow down, or a teleportation). This contrasts with a control condition that shows two identical objects at the end.

- **Block:** Based on Spelke et al. [1992] and Stahl and Feigenson [2015], this scenario tests for notions of solidity and continuity by allowing moving objects to appear on the other side of a solid wall (seemingly having teleported, or moved through the block). In the control conditions, the wall stops the object from going through.

## 5 Experiments

### 5.1 Settings

**Training data.** The training set consists of 1,000 randomly generated videos of objects moving around a scene without collisions (e.g., traveling along a straight line, moving back and forth, rotating; see example videos in the supplementary material). Although ADEPT only uses short sequences of frames to train the perception module, for a fair comparison with baselines we included longer videos with motion patterns typical of objects in the control videos of the test set. Crucially, training videos do not include any physical violations. Objects in these videos can be either occluders, or shapes drawn from 44 ShapeNet [Chang et al., 2015] object categories (the remaining 11 ShapeNet categories are held out for test). Within each category we select only 80% of shapes for the training set, holding out the rest for testing. The training data is used to train the perception module of ADEPT, and all baselines.

**Testing data.** The test set contains 1,512 videos across the eight stimulus classes described in Section 4.2, and are labeled as 'surprise' or 'control' based on the stimulus class design. These videos were designed to have well-controlled configurations of objects and motions that were distinct from the training set. Non-occluder objects in these scenes were never observed in the training set, and come from one of four different shape classes: (1) the training-set ShapeNet object categories, but one of the 20% of shape instances not trained on; (2) the 11 unseen ShapeNet object categories; (3) four simple geometric shapes (cubes, cylinders, cones, spheres); (4) four toy-like shapes similar to those used in developmental experiments (bunny, bowling pin, truck, boot).

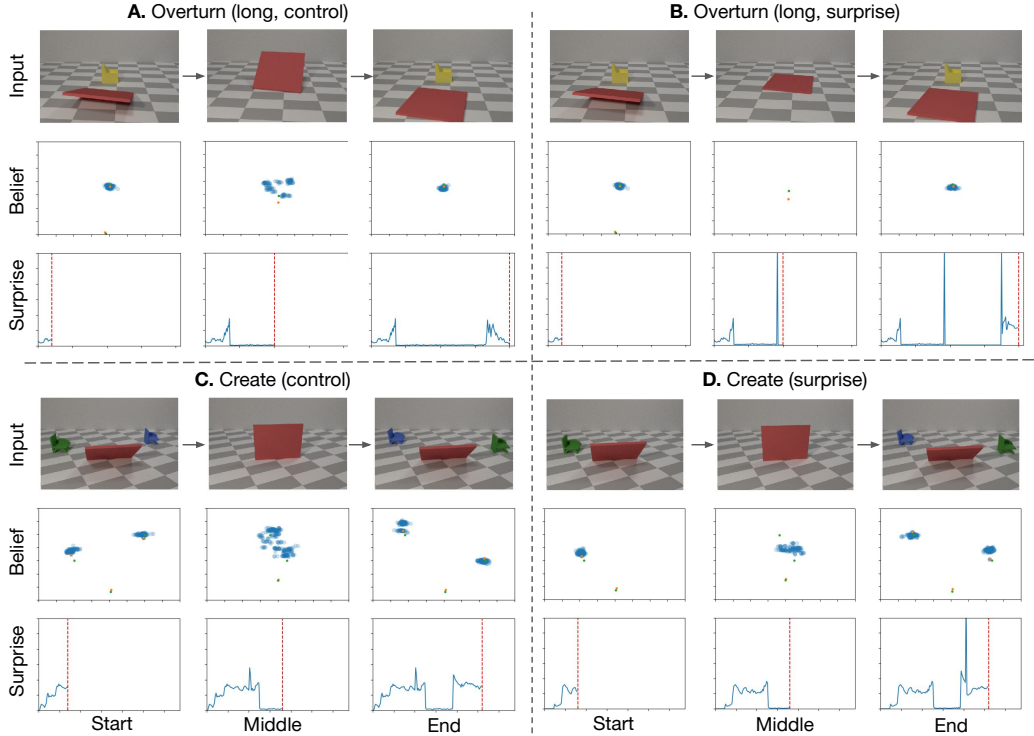

Figure 4: Model performance on four trials. In each block, there are video frames at different moments (row 1), corresponding object location beliefs (row 2) and surprise traces (row 3, the red line indicates the current time). **A.** In *Overturn (long): Control*, the yellow cube is still believed to exist when the occluder overturns (but not completely). **B.** In *Overturn (long): Surprise*, the yellow cube is believed to disappear when the occluder overturns (completely) and then reappear when the occluder overturns back, generating two spikes of surprise. **C.** In *Create: Control*, objects are tracked appropriately and there are no spikes of surprise. **D.** In *Create: Surprise*, when a blue object suddenly appears, a spike of surprise is generated.

**Baselines.** We contrast ADEPT with several standard deep learning models. The encoder-decoder and GAN baselines [Riochet et al., 2018] sequentially take frame pairs $(x_{t-s1}, x_{t-s_2})$ to predict the semantic mask of $x_t$, with a prediction span of 5 and 40, respectively. We have also implemented an LSTM model that explicitly incorporates memory for prediction.

**Metric.** Similar to Riochet et al. [2018], we use relative accuracy to evaluate both model and human judgments. Given a paired group of $n_s$ surprising videos $\{x_i^+\}_{i=1}^{n_s}$ and $n_c$ control videos $\{x_i^-\}_{j=1}^{n_c}$, the relative accuracy within the group is defined as the proportion of surprise-control pairs that are correctly ordered, i.e., $\sum_{i,j} \mathbf{1}\left[c(x_i^+) > c(x_j^-)\right]/(n_s n_c)$. The relative accuracy of several video groups is the average relative accuracy across groups.

Further details on training data, testing data, and baselines can be found in the supplementary material.

## 5.2 Model Results

We present the result of ADEPT and baselines on the entire testing set, which can be grouped into different physical violation cases or shape classes.

**Quantitative results.** Table 1 shows ADEPT outperforms baselines overall. It provides the best performance on four of the eight stimulus classes, and ties for best on three of the four remaining classes. It is also the only model that performs above chance in all cases.

**Generalization to unseen shapes and categories.** All objects in the test set were instances that were not present in the training set, allowing us to test for generalization to novel objects. As can be seen in Table 1, the ADEPT model outperforms or ties with baselines across all object classes, suggesting that it can gracefully handle novel objects. This generalization is driven by the approximate perception module, which was designed to ignore most fine shape information and therefore avoid reliance on previously observed shapes.

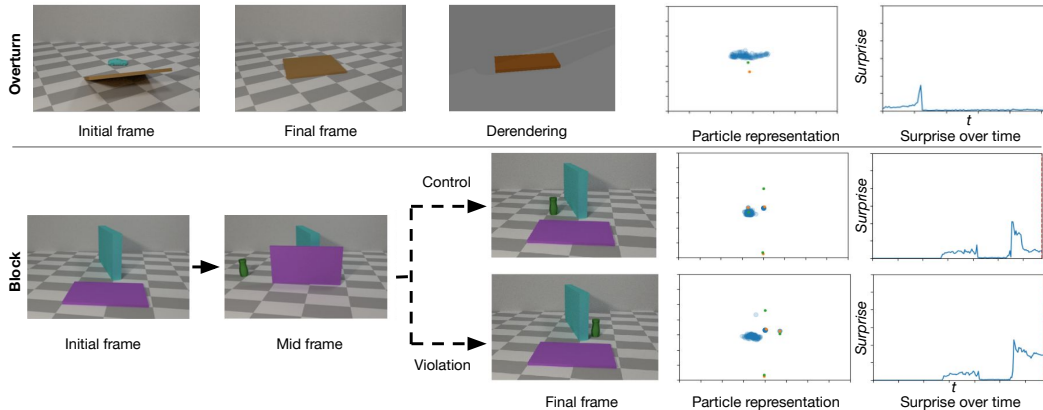

Figure 5: Failure cases in the ADEPT model. *Top:* The scene is initialized with a small object positioned behind an occluder, and the occluder rotates through that object. However, ADEPT perceives the occluder to be only partially rotated, so believes that the object remains underneath and is unsurprised. *Bottom:* An occluder rotates up to partially cover a blocking wall while an object moves from the left into occlusion. In the control case the occluder comes down to show the object on the left side of the wall; ADEPT correctly recognizes this. In the violation case, the object is on the right side of the wall. The perception module localizes the object correctly, but the reasoning module fails to update its beliefs about the object location because there is only a small mismatch in distance.

**Qualitative visualization.** Fig 4 shows the evolution of the model's beliefs (over object location) and surprise, for four videos from two stimulus classes (one surprising and one control from each). ADEPT is highly surprised around the time when the unexpected event occurs: when the occluder moves through the object (top), or an object appears from behind the occluder (bottom). Conversely, during periods with normal physical dynamics the model demonstrates only a low level of surprise, driven by uncertainty about exact object dynamics. See the supplemental videos for further examples of how ADEPT is surprised over time in additional scenarios.

**Limitations.** We find some cases where the ADEPT model sees plausible scenarios as surprising, or surprising scenes as normal. For instance, in Fig. 5 (*top*), coarse representations can cause the perception module to misperceive the occluder's rotation, which produces an unsurprising representation that the object is lying underneath the occluder. In Fig. 5 (*bottom*), ADEPT observes an object on the right side of the block, but represents that object as remaining on the left; because the observation model measures loss in Euclidean distance regardless of intervening objects, this is perceived as simply a perceptual error.

### 5.3 Human Results

In order to directly and quantitatively compare the different models to human intuitions, we asked people to indicate their surprise levels when watching a subset of 64 test videos. 60 participants from Amazon's Mechanical Turk (using the psiTurk framework [Gureckis et al., 2016]) participated for 15-20 minutes, compensated with $2.50. The 64 clips included eight scenes from each of eight conditions described in section 4.2. These were further selected to use two objects from each of the four stimulus categories (unseen shapes, unseen classes, geometric, or toys). Matching "surprise" and "control" videos were selected for each base scene; each participant rated only one of the two. Participants rated how surprising they found the events in each video, using a continuous slider scale from 0 "not at all surprising" to 100 "extremely surprising", and z-scored within participants to control for differences in how people used the slider. We then calculated the relative accuracy for humans (see Section 5.1: Metric).[†]

To determine how well the models capture human surprise, we compare model performance on the subset of scenarios presented in the human experiment. Because peoples' performance differed by violation scenario ($F(14, 3824) = 26.8, p \approx 0$) but not object type ($F(48, 3776) = 0.81, p = 0.83$), we only compare models and humans by violation condition. In six of eight scenarios our model performed at near-human levels, differing substantially only in the 'block' and 'create' scenarios (see Fig. 6). In contrast, all other models did not exceed chance performance on more than half of the

---

[†]Because relative accuracy requires comparisons between surprise and control scenes within a pair, but participants only observed one video per pair, we must aggregate across participants to calculate this metric.

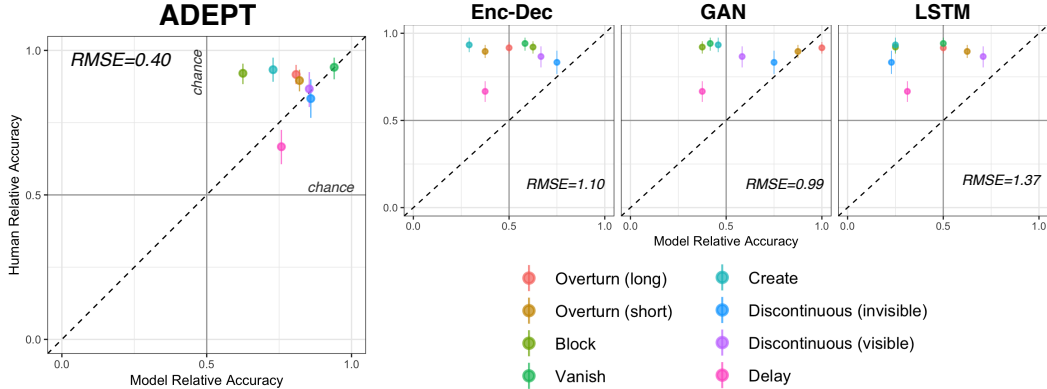

Figure 6: Comparison of model relative accuracy (x-axis) versus human relative accuracy (y-axis) by type of surprise condition for each of the models. Bars represent 95% bootstrapped CIs, the grey lines represent chance performance (50%), and performance is calculated on only the experiment stimulus subset. The ADEPT model performs at near human performance (on the dashed line) in most surprise conditions, and has the lowest deviation from human behavior of all models.

conditions. We quantify this difference by calculating the RMSE between model and human relative accuracies across these conditions, and find that our model has the lowest deviation from human performance by a factor of more than two.

## 5.4 Generalization to other datasets

As a test of whether the ADEPT model generalizes outside of our dataset, we tested it on the IntPhys benchmarks [Riochet et al., 2018] of object permanence (O1) which include more complex textures and motion patterns. In order to capture the texture differences, we retrained the approximate derenderer on IntPhys videos without physical violations, but we retain the same dynamics and observation model used in the experiments above. We find that ADEPT achieves a relative accuracy of 0.73, outperforming all baselines (Enc-Dec: 0.61, GAN: 0.53, LSTM: 0.65).

## 6 Discussion

In this paper, we proposed ADEPT, a model that takes image input and transforms it into an approximate object-based representation, predicts the behavior of objects with probabilistic physical dynamics, and tracks and updates beliefs about those representations. We created a novel Violation of Expectation dataset based on developmental studies, and tested whether ADEPT and pixel-based models hold expectations about objects that are similar to those found in very young infants. We found the ADEPT model best differentiates scenes that violate physical expectations from those that do not, and is the only model that reliably performs above chance across eight different violation scenarios. We additionally measured human surprise on the same violation scenarios, and found that ADEPT matches human performance on many of those scenarios.

Our work highlights the importance of object-centric representations for physical understanding. While visual input might change drastically (due to pose, lighting, or occlusion), object properties tend to remain the same. Holding a core representation of objects can produce more accurate predictions and better generalization.

Finally, we hope this work can help drive a virtuous cycle linking AI and developmental psychology. With models that are surprised by physical events like humans, we can make testable predictions about which physical scenarios people should find surprising, as well as relative surprises across different scenarios. These predictions can in turn drive infant experiments, allowing for model-based tests of the structure and content of infants' object representations. A further understanding of the development of human physical knowledge can then inform the minimal representations and expected learning trajectories needed to design artificial agents that understand physics the way people do.

**Acknowledgments.** This work was sponsored by the Army Research Office accomplished under Grant Number W911NF-18-1-0019, by NSF STC award CCF-1231216, by ONR MURI N00014-13-1-0333, and by Honda's Curious Minded Machines research grant.

## Footnotes

*Indicates equal contribution. Model code, training and testing stimuli, and data can be accessed at `http://physadept.csail.mit.edu/`

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
