[Supplementary Material · SI_Text.pdf]

# Supplementary Material for
# Modeling Expectation Violation in Intuitive Physics with Coarse Probabilistic Object Representations

**Kevin A. Smith**[1,2,*]**, Lingjie Mei**[3,*]**, Shunyu Yao**[4,*]**, Jiajun Wu**[3]
**Elizabeth Spelke**[2,5]**, Joshua B. Tenenbaum**[1,2,3]**, Tomer D. Ullman**[2,5]

[1] MIT BCS, [2] Center for Brains, Minds, & Machines, [3] MIT CSAIL,
[4] Princeton University, [5] Harvard Psychology

## A.1 Stimuli Details

For a better visualization of the test stimuli and how our model handles different violations of expectations in these stimuli, please refer to the additional videos attached in the supplementary material.

**Overturn (long).** This scenario, based on Baillargeon et al. [1985], tests for concepts of solidity and permanence, by showing objects the seem to move through one another. In the "surprise" scene, a screen lies flat on the floor with an object positioned behind it . The screen rotates 180 degrees, initially obscuring the object and then moving through where the object was. The screen finishes lying flat on the floor in the opposite direction to its initial position. After a pause, the screen rotates back to its original position, revealing the object again. The "control" scene starts in the same way, but the screen does not fully rotate, and instead stops before it would overlap with the object. Please refer to SI_Overturn_Long.mp4.

**Overturn (short).** This scenario is identical to the *overturn (long)* scenario, but the video ends after the first rotation of the screen. Please refer to SI_Overturn_Short.mp4.

**Block.** This scenario, based on Spelke et al. [1992] and Stahl and Feigenson [2015], tests for concepts of solidity and continuity. In these videos, a screen starts down with a large, long block positioned just behind it. The screen rotates up 90 degrees to hide the bottom part of the block, then an object starts moving in from the side of the scene. Shortly after the object moves behind the screen, the screen rotates back to its original position. In the "surprise" scene, the object is seen on the opposite side of the block, while in the "control" scene the object is stopped on the same side of the block as it entered the occluded area. Please refer to SI_Block.mp4.

**Vanish.** This scenario, based on Wynn [1992], tests the concept of permanence. In the "surprise" scenario, the video starts with a raised screen, then shows two objects moving from opposite sides of the scene till they are behind the screen. One object then comes out from behind the screen, and the screen is rotated down to reveal that there is no second object behind it. In the three "control" scenarios, either two objects move out from the screen, or only one object moves behind the screen, or when the screen is rotated down an object is revealed to have remained in the occluded area. Please refer to SI_Vanish.mp4.

**Create.** This scenario tests concepts of permanence in the opposite way as the *vanish* scenario, by adding objects to the scene. In the "surprise" scenario, a screen starts down, and then rotates up while one object moves from the side of the scene till it is behind the screen. Two objects then appear from behind the screen, moving in opposite directions. In the "control" scenes, either only one object appears from behind the screen, or two objects move behind the screen and two appear afterwards, or there is a visible object positioned behind the screen initially. Please refer to SI_Create.mp4.

**Discontinuous (invisible).**    This scenario, based on Spelke et al. [1995], tests the principle of continuity, by showing an object disappear behind one occluder and appear out of a spatially distinct region. The "surprise" starts with two narrow screens that have a large gap between them. An object moves out from behind one of the screens towards the edge of the video, then reverses and moves back behind that screen. After a period equal to the time it would take the object to move across the middle space at its last seen velocity, an identical object moves out from the opposite screen towards the scene edge, then reverses and goes back behind that screen. Both screens then rotate down to reveal only an object behind the second screen. In one "control" scene, the video is the same except that when the screen rotates down there are two identical objects, each behind one of the screens. In the other two "control" scenes, the two screens are replaced with one wide screen, with either one or two objects remaining when that screen is rotated down. Please refer to `SI_Discontinuous_Invisible.mp4`.

**Discontinuous (visible).**    This scenario is similar to *discontinuous (invisible)*, but tests concepts of solidity and permanence. These scenes start with the same two narrow screens set apart with a wide gap between them. A first object moves out from behind one screen, then back behind it. However, the first object is then seen visibly moving through the gap between the screens, going behind the second screen, then moving out from behind that screen, and back again behind it. In the "surprise" scene, when the screens rotate down, there are two identical objects, one behind each screen. In one "control" scene, only one object remains behind the second screen. The other two "control" scenes are identical to the *discontinuous (visible)* control scenes with a single, wide screen. Please refer to `SI_Discontinuous_Visible.mp4`.

**Delay.**    This scenario was designed to test the principle of continuity, by showing an object moving through an occluded area too quickly to be explained without teleportation or an abrupt speeding up and slowing down. The scene starts with a single, very wide screen standing up. One object moves out from one side of the screen, then reverses direction and moves back behind the screen. Almost immediately after, an identical object moves out from the other side of the screen and back. The screen then rotates down to the floor. In the "surprise" condition, there is a single object near the side of the screen it was last seen at. In the "control" condition, there are two identical objects, one next to each side of the wide screen. Please refer to `SI_Delay.mp4`.

## A.2   Dataset Generation

To generate videos of physical stimuli, we use the Bullet rigid-body physics engine [Coumans, 2010], and the Blender Cycles rendering engine [Blender]. We produce all images and movies at a $480{\times}320$ resolution. The dimensions of scenes were 8 units by 3 units (width by depth), and all dimensions and velocities reported here are in terms of these units, or units per second. The camera was set at a viewing angle of $20°$. Please refer to `SI_Training.mp4` for sample videos from the training set.

We constructed the training set to have up to one occluder, and up to five objects per scene, moving around for 5 seconds with characteristic motions, but no collisions. Each non-occluder object would have an 80% chance of entering the screen from either the left or right and moving across (with left and right being equally likely), and a 20% chance of starting at a visible location in the screen. Each object would have a 40% chance of reversing motion at a random point in the video. Occluders would always start within the camera's view, and could either rotate fully $180°$, rotate up and down $90°$, or translate back and forth.

Non-occluder objects were scaled to fit within a sphere with a radius uniformly drawn from between 0.2 and 0.4, and moved with a velocity up to 1.5. The width and height of occluders were drawn from a uniform distribution between 1 and 2.

The test set was designed such that object characteristics remained in the same range as training data, though the scenes were much more controlled. Non-occluder objects were scaled to fit within a sphere of radius 0.3, and moved at velocities set depending on their role within the stimulus scenario (between 0.7 and 1.2 when moving). Occluders did not have a characteristic scale, but instead were adjusted based on the stimulus scenario as described above.

## A.3    Model Details

### A.3.1    Perception

For the instance segmentation network, we employ a Mask R-CNN [He et al., 2017] equipped with a ResNet-101 backbone [He et al., 2016] (pretrained on ImageNet) and FPN [Lin et al., 2016]. We train on 125K frames of our 1K training videos, with a batch size of 16, learning rate of 0.008, and 50K total iterations.

For the feature extractor, we use a ResNet-34 [He et al., 2016] (pretrained on ImageNet), in which the first layer and the last two linear layers are resized. We train the feature extractor using an 80% training split of 1,000 train videos, with a batch size of 100, learning rate of 0.002 and 5M total iterations. During training, we evaluate on the other 20% validation split per 5K iterations to choose the best model for inference.

The object state representation for the beliefs ($o_{t,i}$) and for the observations ($s_{t,i}$) is a vector of 22 dimensions to represent object type (background, occluder and non-occluder objects), location (3 entries), velocity (3 entries), rotation (3 entries), scale (3 entries) and color (from 7 colors).

### A.3.2    Dynamics model $p(s_{t+1}|s_t)$

To sample from the dynamics model $p(s_{t+1}|s_t)$, we employ a deterministic rigid body physics engine Bullet [Coumans, 2010]. To create a stochastic physics engine, we perturb the location, velocity, and scale of objects with small Gaussian noises. This simulates the dynamic uncertainty of plausible physics, and also helps to mitigate the perceptual uncertainty (i.e., the error of $o_t$). For our experiment, the noise we use is an additive Gaussian $\sigma_{\text{vel}} = (0.06, 0.01)$ on velocity, $\sigma_{\text{loc}} = (0.06, 0.01)$ on location, and a multiplicative Gaussian of $\lambda_{\text{vel}} = (0.06, 0.01)$ and $\lambda_{\text{dim}} = 0.0005$.

In addition, the dynamics model should be able to reset its beliefs by resampling object properties from a scene prior; if this were not possible, all particles would be rejected upon the first implausible observation, and so the rest of the video could not be parsed. We augment the physics engine with three types of property resampling: with a small probability $p$, an object may suddenly disappear ($p_1 = 0.02$), stop ($p_2 = 0.02$), or speed up in an arbitrary direction ($p_3 = 0.04$). These sampling probabilities are much larger than expected in the scenarios, so that they can potentially be sampled when needed. To mitigate the potential chaos caused by state changes, when one of these changes is sampled, we increase the surprise $-\log p(o_t|s_t^{(m)})$ by a large factor $r$ ($r_1 = 10, r_2 = 1, r_3 = 8$).[*] Thus, a particle undergoing a state change would be resampled if and only if particles undergoing typical physical dynamics are all rejected by the observation, which makes our model robust to physically implausible scenes, but also able to mark them as surprising. This is a form of *importance sampling*: oversampling rare events to consider extreme outcomes when using small hypothesis sets. Importance sampling has been proposed as a way that people consider uncommon events [Lieder et al., 2018], making it a cognitively plausible sampling hypothesis.

### A.3.3    Observation model $p(o_t|s_t)$

To estimate $p(o_t|s_t)$, we first match the objects in our belief $\{s_{t,i}\}_{i=1}^{n(s_t)}$ with the objects in current observation $\{o_{t,i}\}_{i=1}^{n(o_t)}$ based on extrinsic attributes (color, shape, and location). Each $s_{t,i}$ and $o_{t,i}$ can be matched or unmatched, and there are three possible scenarios.

$s_{t,i}$ **matched to** $o_{t,i}$ **for some** $i$.    If the mask produced by belief and the observation has an intersection over union (IoU) above a threshold $thresh_{\text{iou}}$, then we consider that these are the same object for sure. Otherwise, the NLL of $p(o_t|s_t)$ is measured by a combined loss between the belief and observation in terms of location, velocity and scale. The loss metric $\mathcal{L}$ we use is a combined $\mathcal{L}_2$ and $\mathcal{L}_{\frac{1}{2}}$ loss. And the observation model is

$$-\log p\left[o_{t,i}|s_{t,i}\right] = \sum_{\text{attr}\in\{\text{loc,vel,dim}\}} \mathcal{L}((o_{t,i,\text{attr}} - s_{t,i,\text{attr}})/\sigma_{\text{attr}}), \quad \text{if IoU} > t_{\text{iou}},$$

where $\sigma_{\text{attr}}$ is a constant. We use $\sigma_{\text{loc}} = 0.2$, $\sigma_{\text{vel}} = 0.2$, and $\sigma_{\text{dim}} = 0.05$.

---

[*]The model was insensitive to the exact settings of these parameters, so long as this resampling penalty was greater than moderate amounts of perceptual uncertainty.

$s_{t,i}$ **unmatched for some** $i$. We place $s_{t,i}$ in the current observations $o_t$ to see whether the object is hidden by acquiring an estimated visible area $A = A(s_{t,i}, o_t)$. If $A$ is bigger than a visible threshold of $A_0$, we measure the NLL of $p(o_t|s_t)$ based on a quadratic form of $A$. Here, the observation model is

$$- \log p\left[\varnothing | s_{t,i}\right] = \alpha_{\text{base}} + \alpha_{\text{hidden}} A^2,$$

where $\varnothing$ indicates the event that the object belief is unmatched, $\alpha_{\text{base}} = 1$, $\alpha_{\text{hidden}} = 2.5 \times 10^{-5}$, $A_0 = 200$.

$o_{t,i}$ **unmatched for some** $i$. We retrospectively reaffirm if $o_{t,i}$ obeys in the previous history by feeding it through the physics simulation reversely: $\Theta^{-j}(o_t, i) = o'_{t-j,i}$. We then determine its likelihood through the aggregated visible mask area. The reverse physics simulation only permits the 'stop' state change, which allows an object to start moving behind an occluder. Here the observation model is

$$- \log p\left[o_{t,i} | \varnothing\right] = -\alpha_{\text{new}} \sum_{j=t_0}^{t} A_j, \quad \text{where } A_j = A(o'_{t-j,i}, o_{t-j}),$$

where $\alpha_{\text{new}} = 0.02$ and $t_0 = 15$.

We take the average across all these results of $p\left(o_{t,i} | s_{t,i}\right)$ as an estimate for $p\left(o_t | s_t\right)$.

## A.4 Experiment Details

### A.4.1 Details of baseline models

**CNN encoder-decoder.** We use a ResNet-18 [He et al., 2016] pretrained on ImageNet [Deng et al., 2009] to encode features from $x_{t-5}$, a deconvolution network to decode $\hat{m}_t$, and $||\hat{m}_t - m_t||_2^2$ as the training loss. We also train a semantic mask predictor $x_t \rightarrow \hat{m}_t^{MP}$ with the same architecture. Then during testing, $c(x, t) \triangleq ||\hat{m}_t - \hat{m}_t^{MP}||_2^2$ is used as the level of surprise at time $t$.

**GAN.** We first turn input $x_{t-40}$ into $\hat{m}_{t-40}^{MP}$ via the semantic mask predictor. Conditional on these predicted masks, the generator $G$ is trained to generate $\hat{m}_t$, and the discriminator $D$ is trained to distinguish between $\hat{m}_t^{MP}$ (real) and $\hat{m}_t$ (fake). During testing, $c(x, t) \triangleq 1 - D(\hat{m}_t | \hat{m}_{t-40}^{MP})$ is used as the level of surprise at time $t$.

**LSTM.** We use the encoder head of the mask predictor as a feature extractor $x_t \rightarrow z_t^{MP}$. The LSTM is trained to predict $\hat{z}_t$ given $z_{1...t-5}^{MP}$. $c(x, t) \triangleq ||\hat{z}_t, z_t^{MP}||_2^2$ is used as training loss, as well as the level of surprise during testing.

All baselines are trained for five epochs using a batch size of 64 and an Adam optimizer. The learning rate is $8 \times 10^{-4}$ for the generator, $2 \times 10^{-4}$ for the discriminator, and $1 \times 10^{-4}$ for all other networks.

### A.4.2 Qualitative visualization of baseline models

We compare ADEPT's surprise signals over time to those produced by the baseline models. Fig. A1 shows these comparisons for the same scenarios displayed in Fig. 4 of the main paper. ADEPT's surprise score is calculated as the negative log-likelihood of an observation, making it unbounded, but is capped here at a value of 12 (representing more than a very surprising resampling event). All baseline model surprises are calculated as normalized L2 divergences between predictions and observations, and so are naturally scaled between 0 and 1.

The ADEPT model surprise differences between the violation and control videos are much larger that the differences produced by the baseline models. Furthermore, ADEPT's surprise spikes at time points that are intuitively surprising to people: when the object is crushed by the screen (Fig. A1B) or a new object appears from behind the occluder (Fig. A1D). In contrast, at the time of intuitive surprise the baseline models often produce very similar surprise signals in the control and violation videos.

### A.4.3 Ablation study on the number of particles

We conduct an ablation study by reducing the number of particles in the reasoning module of ADEPT. We evaluate our model on the human stimuli set with 128 (default), 32, 16, 8, and 4 particles. As

Figure A1: ADEPT vs. all baseline performance on four trials (see Fig. 4 in the main paper for trial details). Each line represents the surprise signal produced by that model at each point in time, up until the frame above it. The ADEPT model surprise spikes at time points that people find intuitively surprising: when the box is crushed (B) or a new object appears (D). Conversely, the baseline models to not substantially differentiate the control from violation videos at these time points.

shown in Fig. A2, relative accuracy improves as we run on more particles, which suggests a decent level of stochasticity is required to model the noisy (or even implausible) dynamics with coarse perception results.

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

Figure A2: Relative accuracy versus number of particles used.

Tsung-Yi Lin, Piotr Dollár, Ross B. Girshick, Kaiming He, Bharath Hariharan, and Serge J. Belongie. Feature pyramid networks for object detection. *CoRR*, abs/1612.03144, 2016. URL http://arxiv.org/abs/1612.03144. 3

Elizabeth S Spelke, Karen Breinlinger, Janet Macomber, and Kristen Jacobson. Origins of knowledge. *Psychol. Rev.*, 99(4):605, 1992. 1

Elizabeth S. Spelke, Roberta Kestenbaum, Daniel J. Simons, and Debra Wein. Spatiotemporal continuity, smoothness of motion and object identity in infancy. *British Journal of Developmental Psychology*, 13(2): 113–142, 1995. ISSN 0261510X. doi: 10.1111/j.2044-835X.1995.tb00669.x. 2

Aimee E Stahl and Lisa Feigenson. Observing the unexpected enhances infants' learning and exploration. *Sci.*, 348(6230):91–94, 2015. 1

Karen Wynn. Addition and subtraction by human infants. *Nature*, 358(6389):749–750, 1992. ISSN 0028-836, 1476-4687. doi: 10.1038/358749a0. 1