[Reviews · NeurIPS 2019]

Reviewer 1



Many studies have looked at the ideas of physics simulation as a cognitive model. In such works, physics engines are usually employed as a model of human cognition of physical tasks, with the perception part of the task is often abstracted away. In parallel, data driven model have been frequently used to learn to parse raw visual inputs to detect or locate objects, frequently without using any explicit model of the physical world. This paper tries to bridge these two fields to build a complete model of how humans perceive certain physical scenarios, from raw pixels to expectations over objects. Whereas all of the parts employed in the proposed "pipeline" are based on previous works, their arrangement into this contiguous framework is new, as is the human and modeled results on the new dataset the authors also present. The paper is clear and well written in general. However, I do think it would benefit by making its goal clearer. Is the purpose of the proposed framework to be a model of human cognition (for example, the results with ellipsoids might suggest that humans might be performing some sort of simple shape processing at early stages of visual processing)? Or is it to propose a possible new route for improvements in computer vision? (And if so, what is the path the authors see for this type of architecture to contribute to better visual models? Or is surprise detection intrinsically valuable?) Some further clarifications that would be useful: - How fast does the whole pipeline run? How does it compare to the baselines used? - Are the results in Figure 5 different if you calculate accuracy per-person and then average the accuracies? (Instead of averaging scores across people and then calculating accuracies.) - Are there qualitative visualizations (as in Figure 4) for the predictions of the baseline models? Could be interesting in the supplement, if the results are not completely trivial - The GAN model uses a surprise score based on the discriminator. Are the results similar if a L2 loss like the one for the LSTM is used? - Isn't it a bit ad hoc that the values of the surprise for the "rare" events are hardcoded at values seemingly unrelated to their already hand-picked probabilities?(Supplement line 113, r values.) Any specific reason for these values? In sum, this paper presents an interesting combination of object detection with data-driven models, and probabilistic dynamics modeling to construct a complete pipeline for implausibility detection. Though the type of tasks employed are somewhat specific, and modeling surprise directly does not seem to have immediate practical applications, the framework works points in an interesting direction of more integrated modeling and presents interesting comparison to human results. The paper would probably benefit from more explicit discussion of its modeling results in comparison to the human data, and the possible inferences about human cognition.

Reviewer 2



To summarize, the authors present a model to discriminate physically plausible from implausible scenes in occlusion based intuitive physics setups. Images form the input to a visual derenderer the decomposes the scene into objects with physical and deliberately coarse visual attributes to facilitate generalization. The inferred physical object states are fed into a non-learned physics engine (Bullet) which predicts future physical object states. Multiple future beliefs are generated by repeatedly perturbing the input and rerunning the physics engine. A particle filter is then used to combine, track and update beliefs about future object states. Finally the beliefs are compared against actual observations to generate a measure of surprise of the system. Experiments on expectation violation setups based on classical developmental psychology experiments show that the model is reliably able to distinguish physically plausible from implausible scenes and that its predictions align with human predictions about physical plausibility. Originality: Visual object-centric derenderers, particle filters and expectation violation datasets have been proposed before. Those pieces have been put together in a nice way though. Quality: Very well written, clearly organized paper with nicely executed experiments and convincing results. Clarity: Overall the paper is clear and well written. However, all limitations and failure cases have been put into the supplement which is deceiving to the reader and those should be moved to the main paper. Significance: The results are significant to the cognitive science community, but the significance of this methods in a more general sense for other tasks is questionable as several architectural pieces have been optimized to only work on the presented setup. In its current state, this paper is a borderline weak reject for me, but I would like to accept this paper if my concerns are addressed (see improvements section). Detailed comment: (116-117) How important is this distinction and why do occluders need to be modeled separately from other objects? What happens if you don’t do this? EDIT - After reading the author's response, most of my concerns have been addressed and I am changing my rating from weak reject (score of 5) to a weak accept (score of 6).

Reviewer 3



I think this is a high-quality paper. The model proposed is fairly straightforward, and it is unclear whether any decision made represents a particular engineering contribution in which the model was refined by tests on these datasets, but it is cognitively-motivated at many steps. I really appreciate the train-test split design meant to better mimic human subject experiments -- pushing past standards of how we validate in machine learning is a very useful thing to do. The baselines are useful comparisons and I have no sense that these are weak. I do wish that I had a better sense of whether the surprise metrics on the baselines are reasonable -- looking at the supplementary, it was less than clear for me whether there might be some fairer surprise comparison for any of these. The human subject comparison is careful and a very welcome contribution. From an originality standpoint, the model builds on a variety of object-centric forward models, but it is clearly different than these in a way that lends it to the sorts of expectation violation experiments that it tests. I should emphasize that I consider this train-test split design to be quite novel as well -- some recent works have moved towards a more dev psych=inspired train/test split, but it is far from the norm and I think a very welcome addition. The paper is very clear, with very good structure that allows readers to delve into details at different levels. I quickly knew where everything was and could refer back quickly when I needed to. Put together, I think this is a contribution of some significance. Object-centric representations have matured over the past several years, but this is the first example to my knowledge that actually follows through with an expectation-violation comparison test (with a great train-test split!) like those that have inspired these sorts of models. It contributes to this virtuous cycle between AI and dev psych in a clear way.

[Author Response · NeurIPS 2019]

We thank all the reviewers for their insightful and constructive comments, and will revise the paper accordingly.

**(R1) Goal**. Our work has two aims. Our primary goal is to formalize human (especially infant) physical cognition, with
models that can be tested by developmental psychologists. But we also aim to show how modeling infant cognition can
inspire more robust AI vision systems that extract physical object representations from video and can detect violations
of physical expectations to use as learning signals.[*]

**(R1) Comparative run-times**. On a standard CPU server, baselines run at 2–4s/frame, while ADEPT runs at 6–
7s/frame. We currently have not optimized ADEPT for run-time performance and believe there is ample room for future
improvement, such as GPU acceleration via differentiable particle filtering.[†]

**(R1) Accuracy per-person**. Calculating relative accuracy requires a comparison between surprise/control pairs, but
each participant only observed one video from a pair (to avoid biases from seeing earlier near-identical scenes). We
therefore must aggregate across participants to provide this distribution.

**(R1) L2 loss for the GAN model**. Using an L2-based surprise score, the GAN model has a relative accuracy of 0.41
on our dataset (vs. 0.63 with a discriminator-based surprise score, as shown in Table 1).

**(R1) Baseline model images like Fig. 4**. While baselines do not have
interpretable internal belief states similar to the middle panels of Fig. 4,
we can plot model surprise over time. For example, to the right we see the
GAN model is not additionally surprised when the occluder fully rotates
over the object (vs. ADEPT in which surprise spikes; Fig. 4b).

Overturn (long, surprise), GAN baseline

Input — Surprise — Start — Middle — End

**(R1, R2) Cognitive process underlying human surprise.** ADEPT re-
flects a plausible hypothesis that people have a probabilistic, object-based
model of intuitive physics (e.g., Battaglia et al., 2013[‡]), and that surprise is
driven by low probability events under that model (e.g., Teglas et al.[§]). We plan to further explore the human/model
match in future work, e.g., examining moment-by-moment surprise or neural correlates of object disappearance.

**(R1, R2) ADEPT hyperparameters**. We consider the difference between the sampling probabilities for surprising
events and their losses a form of Importance Sampling that allows rare events to be captured with small, cognitively-
plausible hypothesis sets by sampling those events more often than they should occur (see Lieder et al.[¶] on this
phenomenon in humans). Empirically, we also found that performance was insensitive to the exact values or tracking
methods, so long as physics violations are more surprising than moderate amounts of perceptual uncertainty.

**(R2) Problem setup and generalization.** We designed our model to match objects based on general principles (e.g.,
color, shape, and size constancy). We stress that ADEPT's training was not specific to the test dataset: there were no
unphysical scenes or violations in the training set. The derenderer is trained on sequences of three frames to infer the
instant velocity of objects, not long-term motion patterns. The training set had motion patterns similar to the test videos
to allow fair comparisons with baseline models, which do require longer sequences of motion to form predictions. We
will clarify these points in revision.

**(R2) Generalization to other datasets: IntPhys**. We have run ADEPT on the IntPhys dataset, only retraining the
derenderer to handle different visual object properties. Because the IntPhys test server is offline, we evaluate models on
the validation set of scenes designed to test "object permanence," as described in Riochet et al. This set contains 90
matched sets of videos (2 plausible, 2 implausible within each set). ADEPT achieves an overall relative accuracy of
0.73, outperforming all baselines (Enc-Dec: 0.61, GAN: 0.53, LSTM: 0.65). As R2 noted, the videos of IntPhys have
different visual and motion patterns (e.g., complex textures and gravitational motion), so the high performance on this
second dataset suggests our model generalizes to situations where we have no control over the training and test data.

**(R2) Failure cases**. We agree and will move discussion of failure cases from supplemental to main text in the revision.

**(R2) Dataset**. We will release the dataset along with all code, human data, and model evaluations upon publication.

**(R2) Occluder modeling**. Our physics engine assumes motion changes require force, and so cannot capture the
up-and-down motion of occluders. We chose to model them separately to avoid producing a constant surprise signal.

**(R3) Additional information on metrics and fair comparisons**. We follow IntPhys by using the 'relative accuracy'
metric to compare plausible and implausible videos within each matched set, and also follow their choice of plausibility
metrics for each baseline model. We think relative accuracy is the metric most well matched to the developmental
literature, which compares reactions to surprising scenes directly to those in matched controls.

---

[*]Stahl, Feigenson. Observing the unexpected enhances infants' learning and exploration. Science 2015
[†]Jonschkowski et al. Differentiable particle filters: End-to-end learning with algorithmic priors. RSS 2018
[‡]Battaglia et al. Simulation as an engine of physical scene understanding. PNAS 2013
[§]Teglas et al. Pure reasoning in 12-month-old infants as probabilistic inference. Science 2011
[¶]Lieder et al. Over-representation of extreme events in decision making reflects rational use of cognitive resources. Psych. Rev. 2018

[Meta-Review · NeurIPS 2019]

This is a nice paper that combines several existing pieces together in a useful way. The reviewers largely agree on its interest, while pointing out various issues that should be considered. I think it is well worth including, though I would love to see more justification of the adult human methods in light of the infant human motivations.